# Differential Expression of Immune Genes in the *Rhipicephalus microplus* Gut in Response to *Theileria equi* Infection

**DOI:** 10.3390/pathogens11121478

**Published:** 2022-12-06

**Authors:** Patrícia Gonzaga Paulino, Maristela Peckle, Leo Paulis Mendonça, Carlos Luiz Massard, Sandra Antunes, Joana Couto, Ana Domingos, Daniel da Silva Guedes Junior, Alejandro Cabezas-Cruz, Huarrisson Azevedo Santos

**Affiliations:** 1Department of Epidemiology and Public Health, Federal Rural University of Rio de Janeiro (UFRRJ), Seropedica 23890-000, Brazil; 2Department of Animal Parasitology, Federal Rural University of Rio de Janeiro (UFRRJ), Seropedica 23890-000, Brazil; 3Global Health and Tropical Medicine, Instituto de Higiene e Medicina Tropical, Universidade Nova de Lisboa, 1349-008 Lisboa, Portugal; 4Vaccine Complex, Quality Department—Biomolecular and Immunocytochemistry Tests Section, FIOCRUZ, Rio de Janeiro 21040-900, Brazil; 5UMR BIPAR INRAE-ANSES, Ecole Nationale Vétérinaire d’Alfort, Université Paris-Est, 94700 Maisons-Alfort, France

**Keywords:** immunity, signaling pathways, parasite–vector relationship, horses, tick

## Abstract

*Rhipicephalus microplus* is the only tick species known to serve as a biological vector of *Theileria equi* for horses and other equids in Brazil. The protozoan *T. equi* is one of the causal agents of equine piroplasmosis, a major threat in horse breeding systems. Vector competence is closely linked to the pathogens’ ability to evade tick defense mechanisms. However, knowledge of tick immune response against infections by hemoparasites of the *Theileria* genus is scarce. In the present study, the expression of genes involved in immune signaling pathways of *R. microplus* adults’ guts when challenged with a high or low parasitic load of *T. equi* was evaluated. This research demonstrates divergences in the immune gene expression pattern linked to *T. equi* infection in *R. microplus* since the Toll, IMD, and JNK signaling pathways were transcriptionally repressed in the guts of adult ticks infected with *T. equi*. Moreover, the results showed that different infectious doses of *T. equi* induce differential gene expression of key components of immune signaling cascades in *R. microplus* gut, suggesting a link between the intensity of infection and the activation of tick immunity response. The present study adds knowledge to elucidate the gut immune signaling response of *R. microplus* to *T. equi* infection. In addition, the generated data can serve as a basis for further investigations to develop strategies for controlling and preventing equine piroplasmosis.

## 1. Introduction

Equine piroplasmosis (EP) is a disease that affects horses, donkeys, zebras, and mules, and is caused by three protozoan species: *Babesia caballi*, *Theileria equi*, and *Theileria haneyi* [1]. Clinical signs, ranging from inapparent infection and weakness to acute hemolytic anemia, hemoglobinuria, and death, are exhibited in single parasite infections or coinfections [2]. The financial costs of EP treatment are high, especially when parasites of the *Theileria* genus are involved. In this case, the only drug currently available for the chemotherapeutic clearance of *T. equi* from equine blood circulation is imidocarb dipropionate [2]. However, a study evaluating imidocarb’s efficacy in treating *T. equi* and *T. haneyi* showed that this drug is ineffective against *T. haneyi* and fails to consistently eliminate *T. equi* in horses coinfected with *T. haneyi* and *T. equi* [2]. In addition to the lack of effective treatments for EP, a confirmed diagnosis of this disease prevents equine exports/imports, EP being one of the main impediments to international equine traffic [3].

Equine piroplasmosis is a vector-borne disease transmitted by several tick species of the *Dermacentor*, *Rhipicephalus*, *Hyalomma*, and *Haemaphysalis* genera [4,5]. However, in some countries of South America, including Brazil, the only tick species with vector capacity for the transmission of *T. equi* is the cattle tick *R. microplus* [5,6,7,8]. The capacity of *R. microplus* to transmit *T. equi* has been experimentally confirmed, although the mechanisms that enable this tick to be competent for this pathogen transmission are still unclear [5,6,7,8].

Ticks’ vector capacity is closely related to the immune defense of this arthropod [9]. During the blood meal, the tick ingests the macro- and microgamete stages of the protozoan. Colonizing the tick guts is the first step of the *T. equi* life cycle within the vector, followed by migration to the salivary glands. Subsequently, the parasite is released and transmitted to a vertebrate host. For successful transmission, the pathogen must overcome the immunological barriers present in the tick guts [10]. The innate immunity of arthropod vectors comprises cellular and humoral responses that display specific antimicrobial functions [11]. In ticks, the cellular response is linked to hemocytes, which perform several functions, including phagocytosis, formation of nodules, and encapsulation [12]. Meanwhile, the humoral response in ticks is regulated by signal transduction of four main pathways, which are nuclear factor-kappa B/Toll (NF-κB/Toll), immunodeficiency (IMD), c-Jun N-terminal kinase (JNK), and Janus kinase/signal transducer and activator of transcription (JAK/STAT) [10,12].

Previous studies on the interaction between *R. microplus* and other pathogens, such as *Anaplasma marginale* and *Rickettsia* sp., indicated that gene expression modulation of immune signaling pathway components is critical for the successful colonization and transmission of pathogens [13]. For instance, a study on ticks demonstrated that the Relish (IMD transcription factor) controls *A. marginale* infection in the guts and salivary glands of *R. microplus* [14]. Another example is the silencing of the transcription factor STAT in the salivary gland of *Ixodes scapularis*, leading to a decrease in the production of the 5.3 kDa antimicrobial peptide, which in turn allows greater multiplication of *Anaplasma phagocytophilum* in this organ [15].

The elucidation of the tick immune components involved in the interactions between the vector and the parasite is essential to broaden our understanding of the molecular mechanisms at the tick–pathogen interface. Furthermore, this knowledge is vital in identifying targets for developing new preventive strategies to block the pathogen’s transmission. Thus, the present study aimed to analyze the gene expression of the four main immune signaling pathway components (Figure 1) of the *R. microplus* gut infected with high and low loads of *T. equi.*

## 2. Materials and Methods

### 2.1. Consent of the Ethics Committee

The Animal Use Ethics Committee of the Universidade Rural Federal do Rio de Janeiro approved these procedures (protocol 6486110221).

### 2.2. Experimental Animals

Horses were born in the Universidade Federal Rural do Rio de Janeiro and were kept stabled in individual stalls of the Experimental Station for Parasitological Research W. O. Neitz (UFRRJ) during the experiment. These animals received a diet based on coast cross hay, commercial feed provided three times a day, and water ad libitum.

The positive control was artificially infected when this horse was 2 years old. During this time, the horse was also free from hemoparasites. Subsequently, this horse was splenectomized and inoculated intravenously with cryopreserved *T. equi* strain isolated from a foal of 8 days old at UNESP-FCAV, Jaboticabal, state of São Paulo, Brazil [16]. Since then, it has been stabled in isolated pens and used in experimental studies on interactions between *T. equi* and *R. microplus* [8,17].

The *R. microplus* Porto Alegre strain, maintained in laboratory conditions and free from hemoparasites, was used in the current investigation [18].

### 2.3. Experimental Manipulation of Theileria equi Parasitemia

The present study performed two in vivo experiments (Figure 2). Each experiment included a negative control group with one uninfected horse and a *T. equi*-infected group with one artificially infected *T. equi*-positive horse. In both experiments, the negative horse was a spleen-intact mare of 10 years old, and the positive horse was a splenectomized male of 20 years old artificially infected with *T. equi*. The main difference between experiments 1 and 2 was the different parasitemia levels of the infected horse, which were experimentally manipulated (i.e., increased) in experiment 1. Specifically, before experiment 1 execution, the *T. equi*-infected horse was treated with corticosteroid, which has been reported to increase parasitemia successfully [19,20]. The corticosteroid dexamethasone was only administered to the *T. equi*-infected horse as follows: first day—5 mg/100 kg.bwt i/m, second day—2.5 mg/100 kg.bwt i/m, and third day—1.25 mg/100 kg.bwt i/m. In experiment 2, all procedures were carried out as in experiment 1, except that the *T. equi*-infected horse was not treated with corticosteroid. Parasitemia was measured in whole blood samples collected by punction of the jugular vein from the infected horses of each group. Subsequently, parasitemia was determined by qPCR (see “Detection and quantification of *T. equi* in horses and ticks” below).

### 2.4. DNA Extraction

Horse whole blood samples were submitted for total DNA extraction using a DNeasy blood and tissue commercial kit (Qiagen^®^, Hilden, Germany) according to the manufacturer’s recommendations to determine the parasite load in the blood of horses used in experiments 1 and 2.

### 2.5. Tick Infestation and Collection

For each group of the two experiments, 1 and 2, the tick infestation was performed as follows: 80,000 larvae of the *R. microplus* Porto Alegre strain were placed freely on the horses’ backs. On the 33rd experimental day, the engorged females were recovered from the animals’ bodies.

### 2.6. Tick Dissection

Before dissection, the ticks were immersed in running water, followed by 0.05% sodium hypochlorite solution for 3 min and 70% alcohol for 2 min [21]. Subsequently, the ticks were dried and weighed for homogeneous organization into pools created for gene expression quantification. Tick specimens within each experimental group were arranged into three pools of 15 ticks each. Thus, three biological replicates were included for each infection condition of each experiment.

The ticks were dissected on the same day they were collected from horses’ bodies. The dissection was performed according to an adapted protocol [22]. First, the specimens were fixed one by one in Petri dishes containing black paraffin to increase the contrast. Then, the incision in the marginal sulcus was performed by passing the idiosome, with the aid of fine-tipped ophthalmic scissors and entomological tweezers, under a stereoscopic microscope (LED SMZ 445, Nikon, Tokyo, Japan). After complete incision, the tegument was folded to visualize the internal structures better. Next, the exposed cavity was bathed in a sterile 0.9% saline solution at 4 °C. Finally, the guts were collected and washed in ice-cold 0.9% sterile saline three times before storage in 1.5 mL polypropylene tubes containing 1 mL RNAlater (Invitrogen™, Vilnius, Lithuania) and kept frozen at −80 °C for later RNA extraction.

### 2.7. RNA Extraction and cDNA Synthesis

The RNAlater (Invitrogen™, Vilnius, Lithuania) was removed by centrifugation at 18,000× *g* for 15 min. The pellet with the tick guts was washed three times using 1 mL of phosphate-buffered saline (PBS). RNA of gut pools of ticks fed on *T. equi*-infected and uninfected horses were extracted using an Invitrogen™ TRIzol™ Plus RNA purification kit (Thermo Fisher Scientific, Waltham, MA, USA) according to the manufacturer’s instructions. Total RNA was treated with DNase I (Invitrogen™, Carlsbad, CA, USA) to eliminate contaminating genomic DNA. The quality and integrity of the RNA were evaluated using a NanoDrop spectrophotometer (Thermo Fisher Scientific, Wilmington, DE, USA) and electrophoresis in 1% agarose gel. The RNA was quantified by a Qubit™ fluorometer (Invitrogen™) using the Qubit RNA broad-range assay kit.

Complementary DNA (cDNA) was synthesized with a high-capacity RNA-to-cDNA kit (Applied Biosystems, Vilnius, Lithuania) according to the manufacturer’s instructions. Subsequently, the cDNA was quantified using the Qubit ssDNA assay kit.

### 2.8. Detection and Quantification of Theileria equi in Horses and Ticks

The horse whole blood DNA and tick samples cDNA of the experimental groups were submitted to qPCR reaction [23], which specifically targeted the 18S rRNA sequence from *T. equi*, employing the primer pair Be18SF (5′-GCGGTGTTTCGGTGATT CATA-3′) and Be18SR (5′-TGATAGGTCAGAAACTTGAATGATACATC-3′) and a Be18SP fluorescent probe (5′FAM-AATTAGCGAATCGCATGGCTT-TAMRA3′), labeled at the 5′ end with ReporterDye6-carboxyfluorescein (FAM) and the 3′ end with QuencherDye6-carboxy-tetramethylrhodamine (TAMRA). The reactions were performed in triplicate, with a final reaction volume of 12 μL containing: 1X of TaqMan Universal PCR Master Mix (Applied Biosystems, California, USA), 450 nM of each primer, 250 nM of the probe, and approximately 90 ng of total DNA. All reactions contained a TaqMan exogenous internal positive control reagents VIC probe (Applied Biosystems, CA, USA). The thermocycling conditions were as follows: 50 °C for 2 min, 95 °C for 10 min, and 45 cycles at 95 °C for 20 s, followed by 55 °C for 1 min.

The mean values of the quantification cycles (Cq) obtained from the three biological replicates of the infected group were plotted on a standard curve produced by a previous study [8] to attain the gut parasite load of *R. microplus* obtained from each of the experimental infestations performed.

### 2.9. Gene Expression Assays

Forty genes participating in tick immune signaling pathways were selected for this study (Appendix A). The primers used were designed and reported in previous studies [13,14,17,24]. Before their use, the primers were submitted to an optimization assay to find the best annealing temperature and concentration. Then, a six-point dilution curve was used to verify the efficiency of each primer pair [25].

The gene expression assay was performed using a MicroAmp^®^ Fast 96-well reaction plate of a StepOne Plus thermocycler (Applied Biosystems, Singapore, Singapore). Each qPCR reaction contained 1X Power SYBR Green Master Mix (Applied Biosystems, Warrington, UK), 0.4 µM of each primer, and 45 ng/µL of cDNA in a final volume of 12 μL.

The relative gene expression levels in the control group versus the *T. equi* infected group were normalized with the geometric mean of the Cq values of 40S ribosomal protein S3a (GenBank: XM_037430639) and *β-tubulin* gene (GenBank: CK179480) and were calculated according to the 2-(∆∆Cq) method [26].

### 2.10. Statistical Analysis

The statistical significance of differences observed in *T. equi*-infected and control groups was analyzed using GraphPad Prism, version 8.0 (GraphPad Software, CA, USA). The ∆Cq values were submitted to statistical analysis to investigate the gene expression of significance in the *T. equi*-infected and uninfected groups. First, the ∆Cq values were submitted to the Shapiro–Wilk test to verify normality. Then, according to the previously mentioned test results, the control and *T. equi*-infected groups were compared by the parametric Student’s t-test or nonparametric Mann–Whitney test. *p*-values < 0.05 were considered statistically significant.

## 3. Results

### 3.1. Differential Theileria equi Parasitemia in Infected Horses and Collected Ticks

The positive horse parasitemia resulted in 25,397 ± 2287 parasites/µL of blood, as determined by qPCR [23], which is considered high parasitemia. In the second experiment, the positive horse was not treated with corticosteroids presenting a *T. equi* chronic infection of 155 ± 19 parasites/µL of blood. After dissection, the *R. microplus* gut pools from the first experimental assay showed a high *T. equi* load (85,263 ± 7416 parasites/µL), and in the second experiment, the ticks’ gut pools presented a low *T. equi* load (2803 ± 389 parasites/µL).

### 3.2. Toll Pathway

The *Toll receptor* gene was differentially expressed, being five times upregulated (*p* = 0.004; CI = −3.09–−1.50) in the infected group with a high *T. equi* load when compared to the control group, and the *evolutionarily conserved signaling intermediate protein in the Toll pathways* (*ECSIT*) gene was also differentially expressed, being three times upregulated in the infected group (*p* = 0.03. IC = −3.13–−0.26). The *Toll-interacting protein* (*TOLLIP*) gene demonstrated substantial expression, being 134 times upregulated in the infected group (*p* = 0.002; IC = −9.20–−4.86). The differential expression of other genes related to the humoral immunity of *R. microplus* in response to infection by *T. equi* can be seen in Appendix A.

The exacerbated expression level of the gene coding for the TOLLIP inhibitor protein suggests suppression of PELLE kinase, which showed slightly decreased expression levels (Fold change mean = −0.27) though without statistical significance (Figure 3A). The *TUBE* adapter, the *TNF receptor-associated factor* (*TRAF*), and *Cactin* genes exhibited positive regulation with a high differential expression level without statistical significance. Although the TOLLIP inhibitor had a high expression level in the *T. equi*-infected ticks, signal transduction appeared to follow up to *ECSIT*, as *ECSIT* was differentially expressed with statistical significance (Figure 3A). The following components of the NF-κB/Toll pathway did not demonstrate significant differential gene expression.

In contrast, in the second experiment (low parasite load), the expression levels of the NF-κB/Toll pathway members were not marked and did not show statistical significance (Figure 3B). Furthermore, the *Toll* (*Toll and Toll 18 Wheeler*) receptors and *TOLLIP* genes did not differ in expression levels, suggesting that the low parasite load did not stimulate this pathway activation, as occurred in the first experiment with the high *T. equi* load in the *R. microplus*.

### 3.3. IMD Pathway

Concerning the IMD pathway, some components were significantly differentially expressed. The *peptidoglycan recognition protein* (*PGRP*) and proteins involved in the polyubiquitination process, such as an *inhibitor of apoptosis 2* (*IAP2*), *ubiquitin-protein ligase* (*Bendless*), *ubiquitin-protein ligase* (*EFFETE*), and *ubiquitin-conjugating E2 enzyme* (*UEV1a*), did not show significant differential gene expression. However, the *transformative growth factor β-1 kinase* (*TAK1*) showed high levels of expression (*p <* 0.05), being seven times upregulated in the condition infected by *T. equi* (*p*-value = 0.03; IC 95% = −5.01–−0.9). Despite the significant expression of *TAK1*, an inhibitory effect of its activity was also observed since the *E3 ligase Plenty of SH3* (*POSH*) gene was expressed 46 times more in the infected group compared to the control (*p*-value = 0.03; 95% CI = −8.76–−1.64). These data suggest the transcriptional repression of the NF-κB/Toll, JNK, and IMD signaling pathways by the POSH activity.

In addition to TAK1 suppression, other inhibitors of the IMD pathway showed high gene expression levels (Figure 3C). The *factor associated with Fas 1*, *Caspar*, was seven times more expressed in the infected condition, although it was not statistically significant (*p*-value = 0.41; 95% CI = −5.39–2.82). Caspar is an inhibitory regulator of the IMD pathway under homeostasis conditions, maintaining Relish in the cytoplasm and preventing the death-related protein ced-3/Nedd2 (Dreed) from cleaving the transcription factor for entry into the nucleus. Although Caspar has been identified in ticks, Dreed’s presence has not yet been reported [13]. While Caspar was not statistically different expressed, the *Caudal* gene has a high expression level, and it was 12 times more expressed in the infected group compared to the control group (*p*-value = 0.03; 95% CI = −5.09–−0.53). Moreover, the *microplusin* gene presented a slight differential expression, with 0.65 more expressed in the infected group (*p*-value = 0.5; 95% CI =−8.211–4.877) (Figure 4). *Ixodidin* was 4.25 times downregulated (*p*-value = 0.82; 95% CI = −5.96–5.16), and *defensin* showed an upregulation level of 10 times more expression in the infected group compared to the control group (Figure 4). However, these antimicrobial peptides did not exhibit statistical significance in these differences in gene expression levels.

Experiment 2 aimed to analyze differential gene expression in *R. microplus* gut presenting a low *T. equi* load, yet no component of the IMD pathway showed statistically significant changes in expression levels (Appendix A), contrary to the results previously obtained analyzing gut samples exhibiting *T. equi’s* high load condition (Figure 3D).

### 3.4. JNK Pathway

In the JNK pathway, differentially expressed genes in response to the *T. equi* infection in the gut of *R. microplus* were not observed. Although *Basket* (*JNK kinase*) gene was four times upregulated in the *T. equi*-infected group, this difference was not statistically significant (Figure 3E,F). However, in the low parasite load condition (Experiment 2), an integrant of the JNK pathway, the *Fos-related antigen/Kayak* (*FRA*) gene, presented significantly reduced expression levels. The *FRA* gene was 2 times less expressed in the infected condition when compared to the control group (*p*-value = 0.04; 95% CI = 0.19–4.83) (Figure 3F).

### 3.5. JAK/STAT Signaling Pathway

Regarding experiment 1, analyzing samples with a high *T. equi* infection load, the JAK/STAT pathway showed only one differentially expressed gene—*Janus kinase* (*JAK*). The *JAK* gene was approximately 12 times more expressed (*p*-value: 0.02; 95% CI: −5.95–−1.19) in the infected group compared to the control group (Figure 3G). However, other components of this pathway, such as *STAT*, the *STAM* adapter, and the *STAT inhibitory protein* (*PIAS*), did not show significant differences in gene expression levels. In Experiment 2 (low parasite load), none of the JAK/STAT pathway components showed significant expression levels (Figure 3H).

## 4. Discussion

Previous investigations comparing signaling pathways in arthropods have shown that participants in the Toll cascade are conserved in ticks, as observed in studies with tick cell lines (BME26) [13]. In addition, differential gene expression of Toll’s components has already been described in *R. microplus* infected with intracellular bacteria (*Anaplasma marginale*, *Rickettsia rickettsii*), Gram-negative bacteria (*Enterobacter cloacae*), Gram-positive bacteria (*Micrococcus luteus*) and yeast (*Saccharomyces cerevisiae*) in BME26 cells [13]. In the present study, the NF-κB/Toll pathway seemed to be stimulated in the gut of *R. microplus* by the *T. equi* presence since some NF-κB/Toll pathway members were differentially expressed in high-load conditions. Furthermore, the *Toll receptor* gene showed significantly increased expression levels in the *T. equi*-infected group, presenting a high parasite load. The overexpression of the *TOLLIP* gene would suggest an interruption of signal continuity since this modulator is responsible for inhibiting PELLE (IRAK) phosphorylation and this kinase activity [27]. However, *ECSIT*, a transducer following PELLE and TNF receptor-associated factor-like (TRAF), is significantly upregulated (three times) in the infected group. This adapter molecule bridges TRAF and the mitogen-activated protein kinase/ERK kinase-1 (MEKK-1), which can activate Rel and AP-1 family transcription factors [28]. This way, signal transduction appears diverted from the NF-κB/Toll pathway.

Manipulations of the NF-κB/Toll pathway by parasites of the *Theileria* genus in leukocytes from vertebrate hosts are already well supported [29]. *Theileria* recruits and phosphorylates IKK signalosome α and β subunits, which further phosphorylates inhibitory κB (IκB), setting NF-κB free to translocate to the nucleus. Thus, this pathway remains active in the parasites’ presence in vertebrates [29]. However, the IκB, Cactus, the Cactus-interacting protein Cactin, and the pathway transcription factor Dorsal did not exhibit significant differential gene expression, suggesting that activation of the NF-κB/Toll pathway does not occur in the guts of adult ticks.

Regarding the IMD pathway, the *PGRP receptor* gene and the enzymes participating in the K-63 polyubiquitination chain process (*Bendless*, *UEV1a*, and *EFFETE*) did not demonstrate significant differential gene expression. However, signal transduction appears to come from the NF-κB/Toll pathway, as the differential expression of *ECSIT* suggests the activation of MEKK-1. In *Drosophila*, ECSIT binds to TRAF and induces the transcription of defense genes [30]. The ECSIT protein works by promoting the interaction of TRAF with MEKK-1 and completing this kinase processing [30]. In turn, MEKK-1 activates the kinases of the IKK complex (IKK-α and IKK-β) [30]. The IKK complex phosphorylates the Relish transcription factor for subsequent translocation to the nucleus [31].

Another strategy for activating the IKK complex is executed by the TAK1 modulator [32]. In the present study, this kinase gene was seven times upregulated (*p* < 0.05) in the gut of *R. microplus* with a high *T. equi* load. The differential expression of *TAK1* can activate both the IMD and JNK pathways [33]. However, in this research, the TAK1 inhibitor, *POSH*, also showed positive levels of differential expression (*p <* 0.05), being relatively more expressed in the high *T. equi* load infected group. The POSH protein is responsible for the ubiquitination of TAK1 and its subsequent degradation [34]. The degradation of TAK1 entails the rapid termination of signal transduction for Relish and AP-1 [35]. Therefore, POSH’s increased expression in the ticks with a high load of *T. equi* evidences the transcriptional repression of the IMD and JNK pathways in this condition.

In addition to the high *POSH* gene expression levels, the IMD pathway is also repressed by the increased expression of the gene encoding the Caudal protein. This protein is described as a *Drosophila* gut-specific repressor of Relish-dependent antimicrobial peptide genes [36]. The Caudal protein has also been reported as a negative regulator of the IMD pathway in *Anopheles gambiae* infected with *Plasmodium falciparum.* Mosquitoes with a silenced caudal gene showed resistance to *P. falciparum* infection [37].

In the *T. equi-*infected condition, the high differential expression of these inhibitors may suggest tick immune response modulation by the protozoan since the repression of the signaling pathways hampers the production of antimicrobial peptides (AMPs). Microplusin is an AMP directly related to the IMD pathway since the expression of *Relish* increases *microplusin’s* gene expression [14]. This AMP did not present differential gene expression in either experiment, which may corroborate the parasite repression hypothesis.

Following the JNK pathway route, besides the differential expression of *TAK1*, MEKKl can also phosphorylate the JNK (Basket) kinase protein. In a high-parasite-load condition, Basket is four times more expressed in the infected group, though without statistical significance (*p*-value = 0.08). Likewise, other JNK components did not present a significant differential gene expression. Contrasting with a frequent feature of leukocytes transformed by *Theileria* parasites that exhibits constant activation of the JNK pathway [38]. This continuous activation was not observed in tick guts infected with *T. equi*. This distinct expression pattern may be linked to the parasite’s evolutionary phases. In tick gut, sexual reproduction occurs, a very different stage from what happens in the vertebrate host since one of the stages of asexual reproduction occurs in the blood. Hence, these evolutionary forms of protozoan can act in different ways to modulate the signaling pathways of their hosts.

In contrast to the JNK pathway, the JAK/STAT cascade shows evidence of stimulation when there is a high parasite load of *T. equi*, as the *Janus kinase* (*JAK*) gene exhibits significantly increased expression levels in this condition. This pathway has already been reported to be activated by bacteria, viruses, or protozoa and to participate positively or negatively against infections in ticks [12]. However, the following components of the JAK/STAT pathway did not show significant differences in gene expression levels. The suppressor of cytokine signaling (SOCS) can influence the continuity of signal transduction by inhibiting the catalytic activity of JAK, which initiates signaling within the cell [39]. Therefore, this inhibitor could interrupt signal transduction by inhibiting JAK if it is differentially expressed at positive levels. Although the *SOCS* isoform tested in the present study did not show significant differences in expression levels, other *SOCS* isoforms may be involved in this repression. In addition, the AMPs *Ixodidin* and *Defensin* did not display significant gene expression differences. Therefore, this pathway may not be activated in the guts of adult ticks in response to *T. equi* infection.

The overall analysis of these two experiments evidenced several genes with significant differences in the expression levels in the groups infected with high loads of *T. equi* (Figure 3A,C,E), suggesting that the significant differential gene expression of the components of the signaling pathway is dependent on the parasite load. Natural infection in immunocompetent horses presents a chronicle character similar to experiment 2 [8]. When the horse shows low parasitemia, the tick ingests low *T. equi* load during blood feeding; consequently, there is little stimulus for the immune response in the gut of *R. microplus*. This lack of tick immune response may favor *T. equi* to complete the biological cycle. Other studies have already confirmed that even when ticks feed from a horse presenting low parasitemia, the tick acquires and develops the complete protozoan life cycle [8,40].

However, by increasing the parasitemia, the authors simulated blood feeding during acute infection in horses (high parasitemia), and the results showed different tick responses concerning parasite load. Furthermore, even in high parasitemia, the ticks’ immune defense presents transcriptional repression of key components in the signaling pathways, which also favors *T. equi* to complete the biological cycle.

## 5. Conclusions

The present data suggest that a high load of *T. equi* transcriptionally stimulates the NF-κB/Toll pathway, and the differential expression of the *Toll receptor* demonstrates this stimulus. However, this signal seems transcriptionally repressed by the increased expression levels of *TOLLIP*. Moreover, *ECSIT* is also differentially expressed, suggesting signal translocation (change in the signal direction conduction). Nevertheless, this hypothesis should be further investigated.

Other signaling pathways appear to be transcriptionally repressed in the gut of the *R. microplus* with a high *T. equi* load since *POSH* is significantly differentially expressed, suggesting signal interruption to the IMD and JNK pathways, despite the significant and differential expression of TAK1.

Another factor that leads us to believe that the IMD pathway is transcriptionally repressed is the downregulation of the *Relish* gene, the cascade’s transcription factor, and the upregulation of the *Caudal* gene (Relish inhibitor) in the *T. equi* high-load condition. Moreover, microplusin, an AMP controlled by this signaling cascade, is not differentially expressed.

By studying the effect of parasite load on tick immune response, it can be suggested that distinct *T. equi* loads affect the tick *R. microplus* gut gene expression differently. It stimulates or represses tick signaling pathways, probably indicating that the tick immune system responds to parasite infection proportionally to the infection level or load.

Although this research’s preliminary results suggest transcriptional repression of immune signaling pathways in the gut of adult ticks infected with *T. equi*, further studies involving proteomics analysis are required to confirm this effect. Moreover, the expression of the signaling pathway components in tick gut immune response can vary throughout the different stages of tick development. Therefore, other investigations targeting ticks’ immune response in distinct stages, tissues, and sex (males) must be performed in the future to compare and obtain the complete scenario of the interaction between *T. equi* and *R. microplus*.

## Figures and Tables

**Figure 1 pathogens-11-01478-f001:**
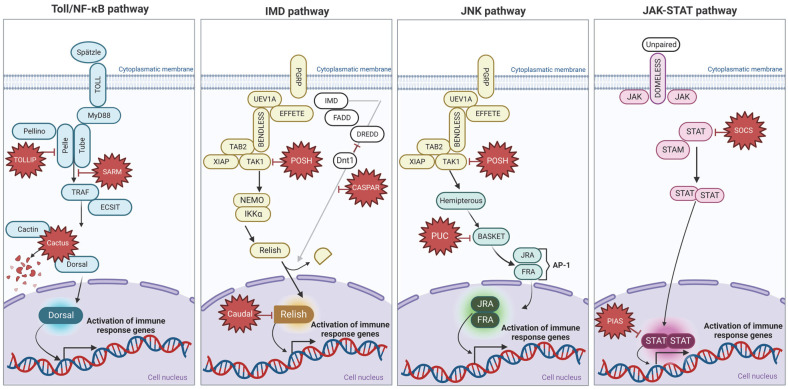
Illustrative representation of the four main signaling pathways that regulate tick immunity: the NF-κB/Toll pathway is indicated in blue, the IMD pathway is indicated in yellow (missing components in ticks of this pathway are shown in white), the JNK pathway is marked in green, and the JAK/STAT pathway is represented in pink. The components in red are inhibitors of the signaling pathways. The figure was created with BioRender.com following the sequence of the components of the signaling route described in Fogaça et al. [10].

**Figure 2 pathogens-11-01478-f002:**
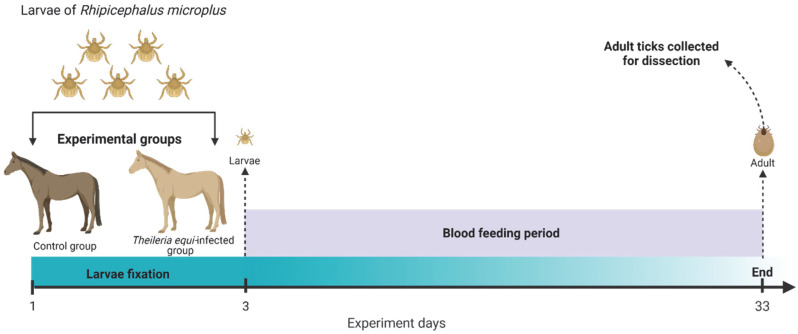
Graphic representation of the in vivo experiments performed in this study. The figure was created with BioRender.com.

**Figure 3 pathogens-11-01478-f003:**
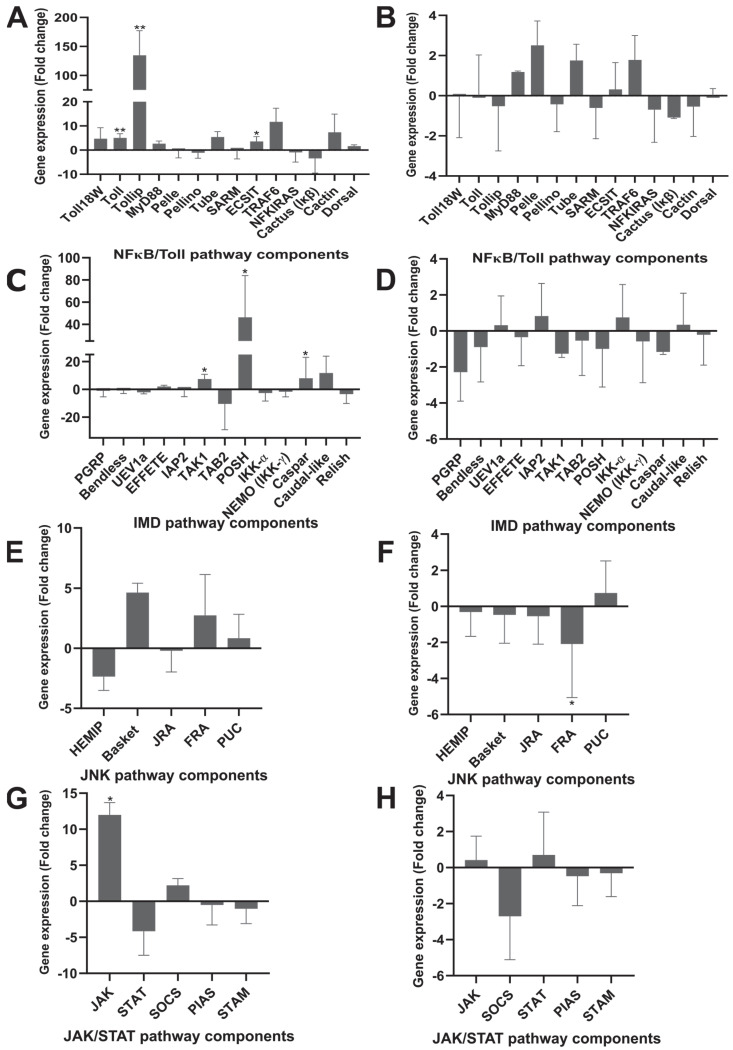
The results shown are means and standard deviation values of gene expression levels. Results in control and *Theileria equi*-infected groups were analyzed by the parametric Student’s t-test or nonparametric Mann–Whitney test. (** *p* < 0.005, * *p <* 0.05; ns: not significant. (**A,C,E,G**) first experiment with high parasite load; (**B,D,F,H**) second experiment with low parasite load).

**Figure 4 pathogens-11-01478-f004:**
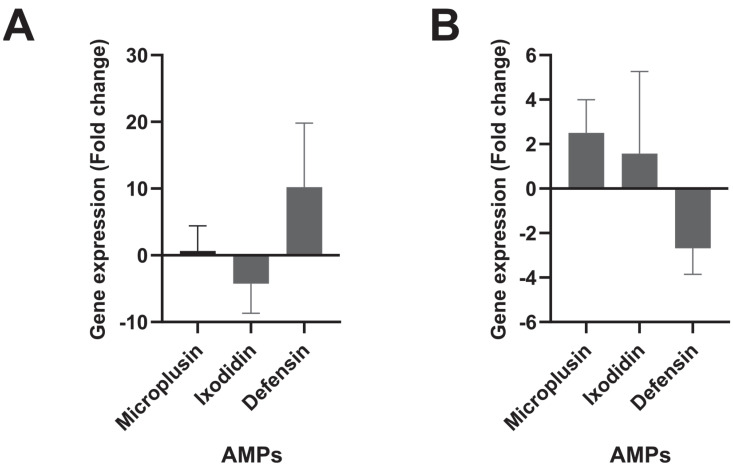
The results shown are means and standard deviation values of the antimicrobial peptides (microplusin, ixodidin, and defensin) gene expression levels in *Rhipicephalus microplus* gut infected by *Theileria equi.* (**A**) First experiment with high parasite load; (**B**) second experiment with low parasite load. Results in control and *T. equi*-infected groups were analyzed by the parametric Student’s t-test or nonparametric Mann–Whitney test.

## Data Availability

Not applicable.

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
