# Peer review of "Differential Expression of Immune Genes in the Rhipicephalus microplus Gut in Response to Theileria equi Infection"

_pathogens, 2022, doi:10.3390/pathogens11121478_

Round 1

Reviewer 1 Report

The manuscript “Differential expression of immune genes in the Rhipicephalus microplus gut in response to Theileria equi infection” by Paulino and colleagues show a panel of all components of the R. microplus Toll, Imd, JNK and JAK/STAT signaling pathways that were possibly modulated by different infection level of T. equi.

My first question is about choosing female ticks. In my view, studying females would be important if Theileria spp. were transovarially transmitted, which is not the case, since this type of transmission does not occur in Theileria species. Furthermore, in engorged females we were not able to dissect salivary glands, which is another important organ to assess the tick's immune response, since the parasite will be transmitted to the vertebrate host during blood feeding, passing through the salivary gland. In this type of study, I would opt for male ticks.

Another point is that the authors evaluated the response in the gut at a single time point, and perhaps for this reason, you have not observed greater gene modulation in more components of the pathways. You can include this in the discussion/conclusion.

I have a few comments below:

Lines 16 and 46-49 - R. microplus is not the only biological vector for T. equi in Brazil. Amblyomma cajennense is also a vector of T. equi. Dermacentor nitens is another tick with an epidemiological association with this parasite, however nothing has been published regarding the competence of D. nitens for T. equi.

Please see: Scoles and Ueti, 2013 - Amblyomma cajennense is an intrastadial biological vector of Theileria equi.

Peckle et al., 2013 - Molecular epidemiology of Theileria equi in horses and their association with possible tick vectors in the state of Rio de Janeiro, Brazil.

Díaz-Sánchez, et al., 2018 - First molecular evidence of Babesia caballi and Theileria equi infections in horses in Cuba.

In Dantas-Torres et al.,2009 - The ticks (Acari: Ixodida: Argasidae, Ixodidae) of Brazil, you can see all the tick species found in Brazil.

Lines 17 and 34 - Equine piroplasmosis affects horses, donkeys, zebras and mules. You can say something like this: ...for horses and other equids in Brazil (line 17) and ... (EP) is a disease that affects horses, donkeys, zebras and mules, caused by... (line 34)

Line 42 - Please remove “-“ in T. haneyi-,

Line 60 - Please provide a reference for cellular responses

Some words should be written in italic: “Rickettsia” (line 66), “Ixodes scapularis” (line 71), “in vivo” (line 80/99), “T. equi” (line 149/188), “Drosophila” (line 324/341), “Plasmodium falciparum” (line 343), Anopheles gambiae” (line 344), “P. falciparum” (line 345)

Line 63 - please remove “[10]. In addition,” The correct is Janus kinase/signal transducer and activator of transcription (JAK/STAT) ...

Line 69/70 - ... in guts and salivary glands of R. microplus

Line 148 - Please change rpm for xg

Line 296-301 - Please include that these results you cite are from an in vitro model, in BME26 cells.

Line 348/349/350 - ...antimicrobial peptides (AMPs). Not PAMPs.

Line 391 - ... level of TOLLIP. Moreover...

Line 401 - ...microplusin, an AMP controlled...

Figure 3 - It has already been shown in silencing experiments of transcription factors of signaling pathways, that Relish regulates the gene expression of AMP microplusin, in R. microplus infected with A. marginale. However, I am not sure if we can say that ixodidin and defensin (as well as lysozyme - which you didn't include in the data) are regulated by Stat (JAK/STAT pathway). Because of this, I would like you to plot the AMPs on a separate graph from the signaling pathways.

Author Response

Reviewer 1

Dear Revisor, we are grateful for your comments and efforts, which were substantially useful for improving the manuscript. The suggestions and inquiries were addressed in the present file and the manuscript.

  1. Reviewer comment: English should be improved. There are lots of long and complicated sentences, which are difficult to read. They should be made brief and clear.

For example, Line 29-30, " generating data that can serve as a basis for further investigations to develop strategies for the control and prevention of equine piroplasmosis."

Line 78-81, "Thus, the present study aimed to analyze the gene expression of components of the four main immune signaling pathways (Figure 1) of the R. microplus gut infected with T. equi in vivo and to verify if the T. equi load influences the gene expression levels of the signaling pathways of the humoral immune response in the gut of R. microplus."

Authors’ answer: Thank you for your suggestion. After the addition of suggestions and corrections the manuscript was sent to English edition.

  1. Reviewer comment: Figure 1. Although the figure is high quality and nice looking, however, it is still similar to the figure in Reference 10. There may be a copyright problem. Furthermore, is it necessary to list it as one of the main figures?

Authors’ answer: Thank you for you comment. The authors agree with you and the figure 1 was altered as follows.

We think is necessary to illustrate to the readers the signaling pathways since some researchers may not be familiar with the position of each component.

  1. Reviewer comment: Line 101, how were the horses naturally infected? You said "they were maintained stabled in individual stalls". Did you just detected the horses in pens and found it infected? It should be made clear.

Authors’ answer: Thank you for pointing out this question. We made a mistake by using ‘naturally’ instead of ‘artificially’. The positive control was artificially infected when this horse was two years old; during this time, the horse was also free from hemoparasites. Subsequently, this horse was splenectomized and inoculated intravenously with cryopreserved T. equi strain isolated from a foal of 8 days of birth at UNESP-FCAV, Jaboticabal, State of São Paulo, Brazil (Baldani et al., 2004). 

Since then, it has been stabled in isolated pens and used in experimental studies on interactions between T. equi and R. microplus (Paulino et al., 2021; Peckle et al., 2022). This horse maintains a chronic and stable infection, where the parasitemia does not fluctuate during the infestation period by R. microplus. For this reason, it was used to study the influence of T. equi load on the immune response of R. microplus.

These informations were added in Material and Methods section, Experimental animals subsection. Lines 97-102 : “The positive control was artificially infected when this horse was two years old; during this time, the horse was also free from hemoparasites. Subsequently, this horse was splenectomized and inoculated intravenously with cryopreserved T. equi strain isolated from a foal of 8 days of birth at UNESP-FCAV, Jaboticabal, State of São Paulo, Brazil [16]. Since then, it has been stabled in isolated pens and used in experimental studies on interactions between T. equi and R. microplus [8;17].”

Baldani, Cristiane Divan, et al. "An enzyme-linked immunosorbent assay for the detection of IgG antibodies against Babesia equi in horses." Ciência Rural 34 (2004): 1525-1529.

Paulino, Patrícia, et al. "Characterization of the Rhipicephalus (Boophilus) microplus Sialotranscriptome Profile in Response to Theileria equi Infection." Pathogens 10.2 (2021): 167.

Peckle, Maristela, et al. "Dynamics of Theileria equi Infection in Rhipicephalus (Boophilus) microplus during the Parasitic Phase in a Chronically Infected Horse." Pathogens 11.5 (2022): 525.

  1. Reviewer comment: The author used corticosteroid to increase parasitemia. However, the parasitemia in animals in the nature can not be such high (just like experiment 2). Actually, the results from experiment 2 may better reflect the real results in the nature. Due to some discrepancies between results of experiment 1 and 2, this should be discussed.

Authors’ answer: Thank you for your comment. This comment was addressed in the discussion section: “The overall analysis of these two experiments evidenced several genes with significant differences in the expression levels in the groups infected with T. equi (Figure 3: A, C, and E), suggesting that the significant differential expression of the components of the signaling pathways is dependent on the parasite load. Natural infection in immunocompetent horses presents a chronicle character similar to experiment 2 [8]. Even when the vertebrate host has low parasitemia, the parasite load acquired by the tick is low; consequently, there is little stimulus for the immune response. This lack of tick immune response may favor T. equi to complete the biological cycle. Other studies have already confirmed that even when ticks feed from a horse presenting low parasitemia, the tick acquires and develops the complete protozoan cycle [8,39].

However, by increasing the parasitemia, the authors simulate blood feeding in acute infection in horses (high parasitemia), and the results show different tick responses concerning parasite load. Furthermore, even in high parasitemia, the ticks' immune defense presents transcriptional repression of key components in the signaling pathways, which also favors T. equi to complete the biological cycle. 

  1. Reviewer comment: Line 71, "Ixodes scapularis" should be italic.

Authors’ answer: The correction was performed as suggested. Lines 70−71: “…and silencing of the transcription factor STAT in the salivary gland of Ixodes scapularis leads to a decrease in the production of the 5.3 kDa antimicrobial peptide…”

Reviewer 2 Report

The work performed by Patrícia Gonzaga Paulino et al., studied differential expression of immune genes in the Rhipicephalus microplus gut in response to Theileria equi infection.
The work is solid, profound, and meaningful for relavant area. However, some revision should be made before publication.

Major comments:
1. English should be improved. There are lots of long and complicated sentences, which are difficult to read. They should be made brief and clear.
For example,  Line 29-30, " generating data that can serve as a basis for further investigations to develop strategies for the control and prevention of equine piroplasmosis."
Line 78-81, "Thus, the present study aimed to analyze the gene expression of components of the four main immune signaling pathways (Figure 1) of the R. microplus gut infected with T. equi in vivo and to verify if the T. equi load influences the gene expression levels of the signaling pathways of the humoral immune response in the gut of R. microplus."
2. Figure 1. Although the figure is high quality and nice looking, however, it is still similar to the figure in Reference 10. There may be a copyright problem. Furthermore, is it necessary to list it as one of the main figures?
3. Line 101, how were the horses natually infected? You said "they were maintained stabled in individual stalls". Did you just detected the horses in pens and found it infected? It should be made clear.
4. The author used corticosteroid to increase parasitemia. However, the parasitemia in animals in the nature can not be such high (just like experiment 2). Actually, the results from experiment 2 may better reflect the real results in the nature. Due to some discrepancies between results of experiment 1 and 2, this should be discussed.

Minor comments
1. Line 71, "Ixodes scapularis" should be italic.

Author Response

Reviewer 2

Dear Revisor, we are grateful for your comments and efforts, which were substantially useful for improving the manuscript. The suggestions and inquiries were addressed in the present file and the manuscript.

  1. Reviewer comment: My first question is about choosing female ticks. In my view, studying females would be important if Theileria were transovarially transmitted, which is not the case, since this type of transmission does not occur in Theileria species. Furthermore, in engorged females we were not able to dissect salivary glands, which is another important organ to assess the tick's immune response, since the parasite will be transmitted to the vertebrate host during blood feeding, passing through the salivary gland. In this type of study, I would opt for male ticks. Another point is that the authors evaluated the response in the gut at a single time point, and perhaps for this reason, you have not observed greater gene modulation in more components of the pathways. You can include this in the discussion/conclusion.

Authors’ answer: Thank you for your comments. This first study aimed at evaluating whether differential immune gene expression modulation occurs in response to T. equi infection. However, the authors agree that other investigations performed in males and tissues in different cycle stages would be interesting to understand better the mechanisms involved in T. equi transmission. Therefore, we have added this comment to our conclusion, as requested.

Lines 416-423: "Although this research's preliminary results suggest transcriptional repression of immune signaling pathways in the gut of adult ticks infected with T. equi, further studies involving proteomics analysis are required to confirm this effect. Moreover, the expression of the components of the signaling pathways in the ticks' gut immune response can vary throughout the different stages of tick development. Therefore, other investigations targeting ticks' immune response in different stages, tissues and sex (males) must be performed in the future to compare and obtain the complete scenario of the interaction between T. equi and R. microplus."

  1. Reviewer comment: Lines 16 and 46-49 - microplus is not the only biological vector for T. equi in Brazil. Amblyomma cajennense is also a vector of T. equi. Dermacentor nitens is another tick with an epidemiological association with this parasite, however nothing has been published regarding the competence of D. nitens for T. equi.

Please see: Scoles and Ueti, 2013 - Amblyomma cajennense is an intrastadial biological vector of Theileria equi.

Peckle et al., 2013 - Molecular epidemiology of Theileria equi in horses and their association with possible tick vectors in the state of Rio de Janeiro, Brazil.

Díaz-Sánchez, et al., 2018 - First molecular evidence of Babesia caballi and Theileria equi infections in horses in Cuba.

In Dantas-Torres et al.,2009 - The ticks (Acari: Ixodida: Argasidae, Ixodidae) of Brazil, you can see all the tick species found in Brazil.

Authors’ answer: Thank you for your comment. Amblyomma cajennense is identified as an intrastadial biological vector of T. equi through experiments conducted by Scoles et al. (2013). However, the taxonomy of the Amblyomma cajennense was redefined after morphological, biological, and molecular studies, currently constituting a complex of six tick species that needs three hosts (trioxene lifecycle) to complete its cycle, infesting mainly equines. Nonetheless, have low host specificity, particularly during the immature stages, and can infest cattle, dogs, birds, and humans (Oliveira, 2000). Therefore, the redefinition of the taxonomy, the species proven by experimental studies with vector competence in the USA, is the Amblyomma mixtum (Scoles & Ueti, 2015). In Brazil, two species of the A. cajennense sensu lato have their occurrence described: A. cajennense sensu stricto (s.s.) and A. sculptum. Although, the occurrence of A. cajennense s.s. is restricted to Northern Brazil (Martins et al., 2016). In contrast, Amblyomma sculptum is a species that occurs in four of the five regions of Brazil (south, southeast, midwest, and northeast) (Nava et al., 2014; Martins et al., 2016).

Amblyomma sculptum and Dermacentor nitens are Brazil's most frequent tick species parasitizing horses (Andreotti, 2019). Based on epidemiological studies, both species were suspected to be T. equi vectors, which reported the statistical association between T. equi presence and A. sculptum or D. nitens infestations (Dos Santos et al., 2011; Heuchert et al., 1999; Keber et al., 2009; Peckle et al., 2013). However, although the geographic distribution of A. sculptum coincides with the areas of occurrence of the disease, it was experimentally verified that the A. sculptum does not have vector capacity for transmission of T. equi (Ribeiro et al., 2011).

Regarding D. nitens, it is still being determined if this tick participates in T. equi transmission.   Other Dermacentor species, such as D. nutalli, D. variabilis, D. marginatus, and D. reticulatus, have experimentally proven vector competence (Battsetseg et al., 2001; Stiller et al., 2002; Ionita et al., 2013). Although, to the present date, no experimental studies investigating D. nitens capacity have been performed in Brazil to support this hypothesis, considered the only tick of the Dermacentor genus present in Brazil (Wise et al., 2013; Andreotti et al., 2019). Moreover, there is a German study in which the researchers report that D. nitens fails to transmit T. equi experimentally (Denning, 1988). However, future studies should be carried out to experimentally investigate the ability of D. nitens to transmit T. equi using more sensitive detection techniques.

Andreotti, Renato, et al. "Carrapatos na cadeia produtiva de bovinos." (2019). Embrapa Gado de Corte -  Livro científico (ALICE).

Battsetseg, Badgar, et al. "Detection of Babesia caballi and Babesia equi in Dermacentor nuttalli adult ticks." International Journal for Parasitology 31.4 (2001): 384-386.

Denning, F. Unsuccessful attempts at transmission of Babesia equi by Anocentor nitens and Amblyomma cajennense. Dissertation - Hannover Veterinary College, 1988.

dos Santos, Tiago Marques dos, et al. "Factors associated to Theileria equi in equids of two microregions from Rio de Janeiro, Brazil." Revista Brasileira de Parasitologia Veterinária 20 (2011): 235-241.Heuchert, C.M.S.; De Giulli, V., Jr.; de Athaide, D.F.; Böse, R.; Friedhoff, K.T. Seroepidemiologic Studies on Babesia equi and Babesia caballi Infections in Brazil. Vet. Parasitol. 1999, 85, 1–11.

Ionita, Mariana, et al. "Molecular evidence for bacterial and protozoan pathogens in hard ticks from Romania." Veterinary parasitology 196.1-2 (2013): 71-76.

Kerber, Claudia E., et al. "Prevalence of equine Piroplasmosis and its association with tick infestation in the State of São Paulo, Brazil." Revista Brasileira de Parasitologia Veterinária 18 (2009): 1-8.

Martins, Thiago F., et al. "Geographical distribution of Amblyomma cajennense (sensu lato) ticks (Parasitiformes: Ixodidae) in Brazil, with description of the nymph of A. cajennense (sensu stricto)." Parasites & Vectors 9.1 (2016): 1-14.

Nava, Santiago, et al. "Reassessment of the taxonomic status of Amblyomma cajennense () with the description of three new species, Amblyomma tonelliae n. sp., Amblyomma interandinum n. sp. and Amblyomma patinoi n. sp., and reinstatement of Amblyomma mixtum, and Amblyomma sculptum (Ixodida: Ixodidae)." Ticks and tick-borne diseases 5.3 (2014): 252-276.

Oliveira, Paulo Roberto, et al. "Population dynamics of the free-living stages of Amblyomma cajennense (Fabricius, 1787)(Acari: Ixodidae) on pastures of Pedro Leopoldo, Minas Gerais State, Brazil." Veterinary parasitology 92.4 (2000): 295-301.

Peckle, Maristela, et al. "Molecular epidemiology of Theileria equi in horses and their association with possible tick vectors in the state of Rio de Janeiro, Brazil." Parasitology research 112.5 (2013): 2017-2025.

Scoles, Glen A., et al. "Equine piroplasmosis associated with Amblyomma cajennense ticks, Texas, USA." Emerging infectious diseases 17.10 (2011): 1903.

Scoles, Glen A., and Massaro W. Ueti. "Amblyomma cajennense is an intrastadial biological vector of Theileria equi." Parasites & vectors 6.1 (2013): 1-9.

Stiller, David, et al. "Dermacentor variabilis and Boophilus microplus (Acari: Ixodidae): experimental vectors of Babesia equi to equids." Journal of medical entomology 39.4 (2002): 667-670.

Ribeiro, Múcio FB, Júlia AG da Silveira, and Camila V. Bastos. "Failure of the Amblyomma cajennense nymph to become infected by Theileria equi after feeding on acute or chronically infected horses." Experimental parasitology 128.4 (2011): 324-327.

Wise, L. N., et al. "Review of equine piroplasmosis." Journal of veterinary internal medicine 27.6 (2013): 1334-1346.

  1. Reviewer comment: Lines 17 and 34 - Equine piroplasmosis affects horses, donkeys, zebras and mules. You can say something like this: ...for horses and other equids in Brazil (line 17) and ... (EP) is a disease that affects horses, donkeys, zebras and mules, caused by... (line 34)

Authors’ answer: The addition was performed as requested. Line 16-17: “Rhipicephalus microplus is the only tick species known to serve as a biological vector of Theileria equi for horses and other equids in Brazil.”

Lines 34-36: “Equine piroplasmosis (EP) is a disease that affects horses, donkeys, zebras and mules, caused by three different protozoan species: Babesia caballi, Theileria equi, and Theileria haneyi [1].”

  1. Reviewer comment: Line 42 - Please remove “-“ in  haneyi-,

Authors’ answer: The removal of the “-“ was performed as requested.

  1. Reviewer comment: Line 60 - Please provide a reference for cellular responses

Authors’ answer: The reference was added as suggested. Line 59-60: “In ticks, the cellular response is linked to hemocytes, which perform several functions, including phagocytosis, formation of nodules, and encapsulation [12].”

  1. Reviewer comment: Some words should be written in italic: “Rickettsia” (line 66), “Ixodes scapularis” (line 71), “in vivo” (line 80/99), “ equi” (line 149/188), “Drosophila” (line 324/341), “Plasmodium falciparum” (line 343), Anopheles gambiae” (line 344), “P. falciparum” (line 345).

Authors’ answer: The manuscript was revised and all genus and species names were written in italic.

  1. Reviewer comment: Line 69/70 - ... in guts and salivary glands of microplus

Authors’ answer: The text was changed as requested. Lines 6ela 8-70: “For example, a study conducted in ticks demonstrated that the relish (IMD transcription factor) controls A. marginale infection in guts and salivary glands of R. microplus [14]…”

  1. Reviewer comment: Line 148 - Please change rpm for xg.

Authors’ answer: The modification was performed as suggested.

Lines 146-147: “The RNAlater (Invitrogen, ThermoFisher) was removed by centrifugation at 18,000 xg for 15 min.”

  1. Reviewer comment: Line 296-301 - Please include that these results you cite are from an in vitro model, in BME26 cells.

Authors’ answer: The inclusion was performed as requested. Lines 395-301: “Previous investigations comparing signaling pathways in arthropods have shown that participants in the Toll cascade are conserved in ticks as observed in studies with tick cell line (BME26) [13]. In addition, differential gene expression of Toll’s components has already been described in infections of R. microplus infected with intracellular bacteria (Anaplasma marginale, Rickettsia rickettsii), gram-negative bacteria (Enterobacter cloacae), gram-positive bacteria (Micrococcus luteus) and yeast (Saccharomyces cerevisiae) in BME26 cells  [13].”

  1. Reviewer comment: Line 348/349/350 - ...antimicrobial peptides (AMPs). Not PAMPs.

Authors’ answer: Thank you for the correction. The alteration was made throughout the manuscript.

Lines 346-351: “In the condition of T. equi infection, the high differential expression of these inhibitors may suggest modulation of the tick’s immune response by the protozoan since the blockade of the signaling pathways inhibits the production of antimicrobial peptides (AMPs). Microplusin is a AMP directly related to the IMD pathway since the expression of relish increases the gene expression of microplusin [14]. This AMP did not present differential gene expression in either experiment, which may corroborate the parasite repression hypothesis.”

  1. Reviewer comment: Line 391 - ... level of TOLLIP. Moreover...

Authors’ answer: The alteration was performed as requested. Line 397-398: “However, this signal seems to be transcriptionally repressed by the high expression levels of TOLLIP.”

  1. Reviewer comment: Line 401 - ...microplusin, an AMP controlled...

Authors’ answer: Thank you for the correction. Line 408: “Moreover, microplusin, an AMP controlled…”

  1. Reviewer comment: Figure 3 - It has already been shown in silencing experiments of transcription factors of signaling pathways, that Relish regulates the gene expression of AMP microplusin, in microplus infected with A. marginale. However, I am not sure if we can say that ixodidin and defensin (as well as lysozyme - which you didn't include in the data) are regulated by Stat (JAK/STAT pathway). Because of this, I would like you to plot the AMPs on a separate graph from the signaling pathways.

Authors’ answer: The suggestion was accepted, and we plotted the AMPs graph (Figure 4) separated from the signaling pathways.

Round 2

Reviewer 2 Report

1. Some sentences are still long and complicated. I suggest to make them short and brief. For example:
line 78-81, "Thus, the present study aims to analyze the gene expression of components of the four main immune signaling pathways (Figure 1) of the R. microplus gut infected with T. equi in vivo and to verify if the T. equi load influences the gene expression levels of the signaling pathways of the humoral immune response in the gut of R. microplus. "
line 355-358, "The tail protein has also been reported as a negative regulator of the immune response via IMD against Plasmodium falciparum in Anopheles gambiae, where mosquitoes with the tail protein gene silenced showed more excellent resistance to P. falciparum in infection [37]."

2. line 396-398, "when the vertebrate host has low parasitemia, the parasite load acquired by the tick is low; consequently, there is little stimulus for the immune response." I am not clear what sentence means. Are you sure it has been English edited?

3. The Discussion section, could you combine some discussion contents according to the four pathway (Toll, IMD, JNK, JAK-STAT)? Some contents on the same pathway can be combined into one paragraph. In my opinion, this will make it more clear.

Author Response

Reviewer 1

Dear Revisor, we are grateful for your suggestion to improve the language, which enhances the manuscript quality. Therefore, we have highlighted your latest question and provided the English edition.

  1. Reviewer comment: line 78-81, "Thus, the present study aims to analyze the gene expression of components of the four main immune signaling pathways (Figure 1) of the microplus gut infected with T. equi in vivo and to verify if the T. equi load influences the gene expression levels of the signaling pathways of the humoral immune response in the gut of R. microplus.”

Authors’ answer: Lines 80-82: "Thus, the present study aims to analyze the gene expression of the four main immune signaling pathways components (Figure 1) of the R. microplus gut infected with high and low loads of T. equi.

  1. Reviewer comment: line 355-358, "The tail protein has also been reported as a negative regulator of the immune response via IMD against Plasmodium falciparum in Anopheles gambiae, where mosquitoes with the tail protein gene silenced showed more resistance to falciparum infection [37]."

Authors’ answer: Lines 355-357: “The Caudal protein has also been reported as a negative regulator of the IMD pathway in Anopheles gambiae infected with Plasmodium falciparum. Mosquitoes with silenced caudal gene showed resistance to P. falciparum infection [37].”

  1. Reviewer comment: line 396-398, "when the vertebrate host has low parasitemia, the parasite load acquired by the tick is low; consequently, there is little stimulus for the immune response." I am not clear what sentence means. Are you sure it has been English edited?

Authors’ answer: Lines 396-398: “When the horse presents low parasitemia, the tick ingests low T. equi load during blood feeding; consequently, there is little stimulus for the immune response in the gut of R. microplus.”

  1. Reviewer comment: The Discussion section, could you combine some discussion contents according to the four pathway (Toll, IMD, JNK, JAK-STAT)? Some contents on the same pathway can be combined into one paragraph. In my opinion, this will make it more clear.

Authors’ answer: We accepted your suggestion and combined some paragraphs to make it more straightforward.